# Feedback ARMA Models versus Bayesian Models towards Securing OpenFlow Controllers for SDNs

Wael Hosny Fouad Aly *, Hassan Kanj , Nour Mostafa and Samer Alabed

College of Engineering and Technology, American University of the Middle East, Egaila 54200, Kuwait; hassan.kanj@aum.edu.kw (H.K.); nour.moustafa@aum.edu.kw (N.M.); samer.al-abed@aum.edu.kw (S.A.)
* Correspondence: wael.aly@aum.edu.kw

**Abstract:** In software-defined networking (SDN), the control layers are moved away from the forwarding switching layers. SDN gives more programmability and flexibility to the controllers. OpenFlow is a protocol that gives access to the forwarding plane of a network switch or router over the SDN network. OpenFlow uses a centralized control of network switches and routers in and SDN environment. Security is of major importance for SDN deployment. Transport layer security (TLS) is used to implement security for OpenFlow. This paper proposed a new technique to improve the security of the OpenFlow controller through modifying the TLS implementation. The proposed model is referred to as the *secured feedback model using autoregressive moving average (ARMA) for SDN* networks (SFBARMA$_{SDN}$). SFBARMA$_{SDN}$ depended on computing the feedback for incoming packets based on ARMA models. Filtering techniques based on ARMA techniques were used to filter the packets and detect malicious packets that needed to be dropped. SFBARMA$_{SDN}$ was compared to two reference models. One reference model was Bayesian-based and the other reference model was the standard OpenFlow.

**Keywords:** software-defined network; network security; OpenFlow controller; ARMA; empirical technique; Bayesian network

## 1. Introduction

Internet softwarization is the future of the Internet and has an impact on network innovation towards network improvement and enhancement. Software-defined networking (SDN) is known for its near-to-optimal solutions regarding network design and implementation. SDN is considered an appropriate environment to become immune to malicious attacks. SDN separates control functionalities and forwarding functionalities. Controller modules are responsible for control decision functionalities, while the switch modules are responsible for forwarding functionalities. A typical SDN architecture is composed of north-bound interfaces, a management layer, a control layer, a data layer, and east–west-bound interfaces. Moreover, the management layer consists of a set of network applications that manage the control logic of the SDN. The control layer contains a set of controllers that forward different rules and policies to the data layer through the southbound interface. The data layer represents the forwarding network elements on the network [1].

SDN architecture has different types of interfaces, such as a northbound interface, an east–west-bound interface, and southbound interfaces. The northbound interface controls the connection between the controller module and the application. It is responsible for the communication between control layer and the management layer. Moreover, the east–west-bound interface allows communication between multiple controllers through message passing. The southbound interfaces facilitate the interaction between the control layer and the data layer [2–5].

OpenFlow is one of the most popular protocols that pushes policies to the forwarding plane. Security of SDN has been under investigation since OpenFlow is considered to be

the first standard communications interface designed by the Open Network Foundation (ONF). OpenFlow security relies on the optional implementation of transport layer security (TLS) that is considered to be vulnerable to malicious attacks. Methods for increasing SDN reliability has been studied in various research works, such as [1,6–10].

This work proposed a novel model to add secure capabilities to SDN networks using feedback control techniques based on ARMA models. The proposed approach was based on feedback control techniques using autoregressive moving average (ARMA) models. The proposed model is referred to as the *secured feedback model using autoregressive moving average (ARMA) for SDN networks* (SFBARMA$_{SDN}$). Extensive comparisons were conducted to compare the proposed model SFBARMA$_{SDN}$ to two reference models. The first reference model is based on Bayesian networks to provide security capabilities and is referred to as the secured standard using Bayesian networks for SDN (SSBN$_{SDN}$). The second reference model is referred to as the standard OpenFlow [11].

The reason behind using the ARMA model in the proposed model was that in classical engineering environments, the relationships between outputs and inputs are studied through physical laws that are referred to as the first-principle techniques. The main barrier for using first-principle modeling in the computing system domain was that some unrealistic assumptions were made. For that reason we used empirical approaches for developing transfer functions through the autoregressive moving average (ARMA) approach [12]. The proposed feedback control system model relied on having a tuning parameter that was easy to control. This tuning parameter had an influence on a controlled output parameter. The system aimed to ultimately achieve a certain target value for the controlled output parameter. The controlled output parameter was a parameter that was needed to be controlled but could not be adjusted directly; hence, the tuning parameter came into place. The tuning parameter could be directly tuned and had an impact on the controlled output parameter. In this work the tuning parameter used was the current value of the security level of the system. The controlled output parameter was the enhanced security level of the system. Feedback controllers used the ratio between outputs and inputs. This was defined mathematically through transfer functions [13].

For the purposes of the SDN server, the level of security attainment was used to characterize the performance of the system. This metric provided a way to manage trade-offs between the achieved security level and the target security level.

Thus, rather than proceeding from first principles, this research used an empirical approach. Obviously, the SDN system in question was highly discrete and nonlinear. However, for the purposes of control, a linear system model was desired. This model was an approximation of the real system at best, but, for control, approximate models often suffice.

The server had many available tuning parameters. Tuning parameters are the parameters that affect the different resources utilized by the system. Some of these tuning controls must be fixed at installation time, while others can be changed on-line while the server is operating. The tuning parameters were considered as the control input in this work and the current value of the security level of the system.

The main reference model used in this work was based on Bayesian networks (BNs). BNs are considered to be probabilistic models that use graphical representations for knowledge. A BN is a domain that deals with uncertain domains. The main strength of a BN is that it applies probability theory to control model complexities. BN nodes are represented as random variables, while edges are represented as conditional probabilities for the appropriate random variables. A major weakness of BNs is the need to fully specify the probability distributions for the network, and this number is high [14].

The paper is organized as follows. Section 2 presents related work. Section 3 describes the SSBN$_{SDN}$ reference model. Section 4 presents the SFBARMA$_{SDN}$ proposed model. Section 5 presents the experimental results. Finally, Section 6 presents the conclusion and future work.

## 2. Related Work

This section briefly discusses the OpenFlow protocol. The OpenFlow protocol is considered as the de facto SDN protocol for SDN. OpenFlow is implemented at both the SDN interface and the SDN controller. It is responsible for forwarding packets in the forwarding plane for all SDN network elements [15]. OpenFlow is categorized into three categories: (1) switch category; (2) channel category; and (3) controller category [16].

The switch has flow tables with flow entry lists. For each arriving packet, the switch matches packets based on the flow table. Selection is performed based on the highest available priority that matches the packet header. In case of similar priorities, the selected flow is not defined. If the packet does not match the row table, then it should be discarded. If the packet does not have a missing flow entry, then the appropriate policies should be followed. OpenFlow channel is an interface that enables controllers to connect to OpenFlow switches. This interface guarantees that the controller makes the following operations: (1) configures the switch; (2) receives various events from the switch; and (3) sends packets to the switch.

Types of messages that are allowed in these channels are controller to switch messages, switch to controller messages to update the controller about network status, and messages by either switches or controllers in emergency cases. In the literature controllers are considered to be either centralized or distributed. An OpenFlow controller is classified as a centralized controller that maintains policies and instructions among network elements. Policies determine the way to manage packets without matching flow entries and also manage the switch flow table by updating flow entries securely. TLS is used as the default security mechanism for the OpenFlow controller since the standard OpenFlow does not provide security [14,15,17–20].

The OpenFlow switch is able to establish a connection and communicate with various controllers. The reliability of the SDN network could be improved through utilizing multiple controllers by resisting failures. OpenFlow switches start by connecting all existing controllers although the messages to be sent should only be sent to the appropriate switch. Several research teams have investigated various security issues for OpenFlow [21–24]. Agborubere et al. [24] proposed a model to improve OpenFlow and TLS communications security. The model summarized the TLS security issues by recommending techniques to improve the TLS. The authors of [24] focused on securing the OpenFlow controller through classifying different types of attacks using a BN classifier for TLS. SDN controllers dynamically add or remove policies and rules. Administrators configure the network and deploy new protocols through the controllers. Therefore, the management of SDN greatly increases the programmability and flexibility of the network.

Meng et al. [14] proposed a management model based on Bayesian models for insider attackers in healthcare environments. Tseng et al. [17] proposed a new model called ControllerSEPA. ControllerSEPA is considered as a lightweight plugin project to protect the network against intruders. Uchupala et al. [8] proposed an application-aware network based on the spanning tree technique in addition to the shortest path routing network for various scenarios. Craiget et al. [23] presented a method for using bloom filters in software-defined networks (SDN) to reduce the state of packet delivery while removing false positive forwarding. Song et al. [18] proposed a technique for an SDN control plane to assign a group of the event processing modules to switches. Authors in this work suggested using switches for dealing with the OpenFlow events rather than the controllers. Qiu et al. [22] suggested using global flow tables for global flow collection, computation of all the paths for network, and also for global flow storing. Xiong et al. [19] proposed a queuing model for analyzing OpenFlow-based SDNs. Silva et al. [21] proposed an extension to the OpenFlow protocol to include flexible time-triggered real-time communication services. The proposed model extended the capabilities of the OpenFlow protocol to support real-time reservations.

### 3. SSBN$_{SDN}$ Reference Model

This section describes the reference model used in this paper. The reference model is referred to as the *secured standard using Bayesian networks for SDN (SSBN$_{SDN}$)* [25]. SSBN$_{SDN}$ has security features implemented into the OpenFlow controller. SSBN$_{SDN}$ uses a Bayesian network model for packet filtering decisions in addition to filtering rules matching techniques. SSBN$_{SDN}$ implements packet filters through utilizing security features of OpenFlow controllers. SSBN$_{SDN}$ inspects packets based on their individual characteristics.

SSBN$_{SDN}$ depends on the process of packet inspection using BN classifiers to determine whether the packet is a malicious packet or not. SSBN$_{SDN}$ performs the filtering process by following the occurrence of the repeated attacks to a specific target with a certain probability. This is used as an indication that the host is being attacked. In this scenario, the policy is to store the attacking packet information in the flow table to be discarded. SSBN$_{SDN}$ uses a Ryu framework for handling OpenFlow since it has conformance with OpenFlow specifications [15].

OpenFlow switches receive a packet on their input ports, and hence matching processes are performed to match the incoming packets to their corresponding entries in the flow table. In the case that the packet does not exist in the flow table, the packet is sent to the controller for more inspection and processing. SSBN$_{SDN}$ uses packet filtering rules to be able to discard malicious packets.

This includes the information found in the packet header, whether a protocol that is encapsulated is being used, source and destination ports, an ICMP message type, and packet ingress and egress interfaces. In case of a packet hit scenario (i.e., matching occurs), the packet is considered as a *green* packet. If it is a packet miss scenario (i.e., no matching), the packet is considered as a *red* packet and, hence, is discarded.

SSBN$_{SDN}$ defines the problem as a set R = $(r_1, r_2, \ldots, r_n)$ of orthogonal instances of the random variables $(R_1, R_2, \ldots, R_n)$. As a result, the network N that best matches R could be found. One of the common approaches used for scoring is the Bayesian score approach. The score works by computing the probability of the data given the directed acyclic graph. Maximizing the Bayesian score using M is an NP-hard problem that could be solved using heuristic techniques. SSBN$_{SDN}$ computes the scores as shown in Equation (1). It represents the probability of the graph.

$$S(N : R) = logP(N|R) = logP(R|N) + logP(N) + M \tag{1}$$

$R|N$ denotes the average data over parametric assignments to N. SSBN$_{SDN}$ comprises a directed acyclic graph (DAG). DAG is composed of a set of nodes, one node is dedicated to each random variable. DAG contains a set of directed edges in additional to a set of conditional probability distributions. Equation (2) defines the conditional probability distribution of each node as a function of its parents.

$$P(X_i|Parents(X_i)) \tag{2}$$

A conditional probability table (CPT) represents a conditional distribution that gives a distribution for $X_i$ for all possible combinations of the parents' values. Intuitively, the conditional probability for Boolean $X_i$ that has k Boolean parents gives up to $2^k$ different possible rows for all the combinations of the parent values. Each possibility requires a probability of P for $X_i$ to be true and a probability of $1 - P$ for $X_i$ to be false. The joint probability distribution coan be represented as shown in Equation (3).

$$P(X_1, \ldots, X_n) = \prod_{i=1}^{n} P(X_i|Parents(X_i)) \tag{3}$$

SSBN$_{SDN}$ utilizes a centralized OpenFlow controller model that establishes a pattern of trust among nodes and also detects untrusted devices for dynamic thresholds. Threshold values are adjusted by the administrator. SSBN$_{SDN}$ classifies the input packets into two classes through filters: safe packets class and unsafe packets. SSBN$_{SDN}$ assumes that if the

probability of an IP destination address is greater than a certain threshold it is considered as an unsafe packet, otherwise it is a safe packet. The security evaluation of SSBN$_{SDN}$ is implemented through Python script to add the necessary security functionalities to the current version of the OpenFlow controller. The network attack workloads were FOI datasets. Three different FOIs datasets were used and results were consistent.

Wireshark was used to monitor inbound and outbound packets. The experiments were conducted using a Mininet emulator version 2.1.1+ installed on an Oracle VirtualBox. A VM was connected to the network using network address translation (NAT). Moreover, the SSH facilitates used the VM to run multiple software packages simultaneously. Three different FOI datasets were used. Results were consistent. The OpenFlow controller security features were applied using two different packet filtering techniques: (1) simple packet filtering techniques that use rules to inspect packet properties; and (2) Bayesian network (BN) filters that observe unusual packet patterns in addition to possible DDoS attacks. BN filtering decisions were performed through keeping track of the repeated attacks on the same host or the same IP destination address with a probability value of 0.8. This was assumed to be an indication of attack occurrence. In such situations, the packet pattern was saved in the flow table to be scheduled for dropping. In this article, the Ryu framework was utilized to handle OpenFlow. The Ryu framework has high conformance with OpenFlow configurations. If it was a hit scenario where the packet matches the filtering rules, it was referred to as a green packet, otherwise it was referred to as a red packet.

A BN model starts by inspecting the incoming packets. If the incoming packet matches permission rules in the flow tables, then the packet routes. If the incoming packet does not match any permission rules from the flow tables, then packet filtering is used. If the packet is a safe packet (green packet), the packet goes to packet filtering using a BN classifier. If the packet is a red packet, the flow tables are updated to be eventually discarded.

## 4. SFBARMA$_{SDN}$ Proposed Model

In this section, the proposed model is discussed. The proposed model is referred to as *secured feedback control using auto regressive moving average for SDN (SFBARMA$_{SDN}$)*. SFBARMA$_{SDN}$ is based on feedback control theoretic techniques. SFBARMA$_{SDN}$ works in two phases. (1) The system identification phase, which is responsible for mathematically modeling the system; and (2) the control law phase that is responsible for detecting the safe and unsafe packet classes based filtering schemes.

Figure 1 depicts the block diagram for the feedback control system utilized by the proposed model SFBARMA$_{SDN}$. SFBARMA$_{SDN}$ added security levels to the SDN through filtering based on ARMA approaches. The output was a higher secure system. After a time slot, the system could became vulnerable to attacks and might return back again to the unsecured state. The observer module detected the level of vulnerability and compared the security level to a threshold value. If the value exceeded the threshold value, then controller's control law took the appropriate action to rectify the malicious behavior.

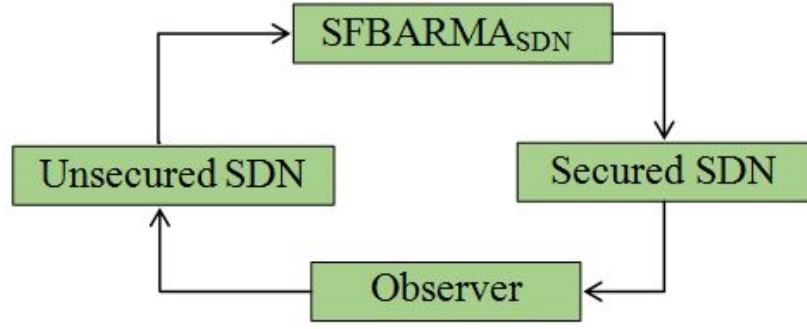

**Figure 1.** Block diagram for the feedback control system for the SFBARMA$_{SDN}$ model.

Figure 2 shows the feedback control system for the SFBARMA$_{SDN}$ model. The *controller module* had a control function that modified the tuning parameter. The *actuator module* was responsible for executing the appropriate actions based on ARMA approaches. The vulnerable system was referred to as the *controlled system module*. The *observer module* was used to detect the current status of the security level of the SDN. The output was fed back to the comparator to be compared to a certain reference value. The reference value was assumed to be 90% in this model.

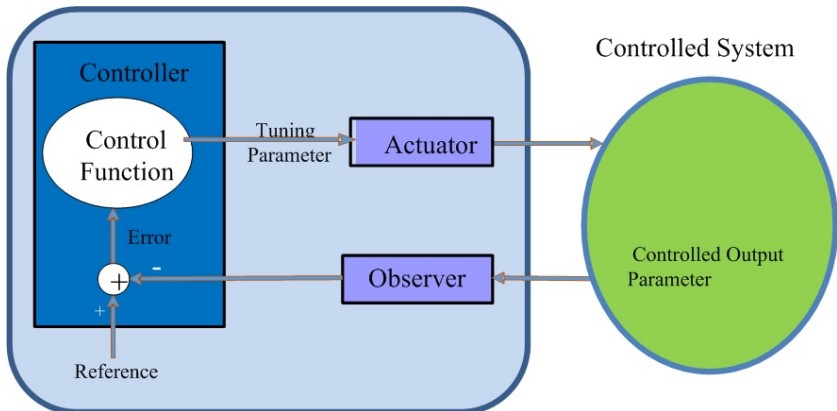

**Figure 2.** Feedback control system for the DFBCP model.

The system identification phase focused on using linear regression to model SDN elements. The generic time domain of the ARMA model is expressed in Equation (4) in terms of the output of the module, $y(t)$, as a function in the input, $x(t)$. The output was expressed as a series of the inputs. ARMA model had $n$ series of the weighted values of the previous output and $m$ weighted values of input values, as shown in Equation (4). The values of $i$ and $j$ were the index values for the previous output and the input values, respectively.

$$y(t) = \sum_{i=1}^{n} a_i \times y(t-i) + \sum_{i=0}^{m} b_j \times x(t-j) \tag{4}$$

In order to ease the mathematical modeling and derivation, a frequency Z-transform version of the ARMA was derived as shown in Equation (5):

$$H(z) = \frac{Y(z)}{X(z)} = \frac{\sum_{j=0}^{m} b_j \times z^{n-j}}{z^n - \left(\sum_{i=1}^{n} a_i \times z^{n-i}\right)} \tag{5}$$

SFBARMA$_{SDN}$ modeled the SDN using a feedback control system, as shown in Figure 3. The SFBARMA$_{SDN}$ model was composed of the security engine that was responsible for maintaining the required security level. The SDN module represented the SDN network that needed to be secured, including the overflow protocol. The observer module represented the sensing element that was responsible for reading the controlled output parameter (COP). SFBARMA$_{SDN}$ assumed that COP was the enhanced security level of the system denoted by $s_i(t)$. The security engine module computed the tuning parameter of the model, which was the current value of the security level of the system denoted by $u_i(t)$. The relationship between $s_i(t)$ and the $u_i(t)$ is given by Equation (6) by applying it into the ARMA mathematical model explained in Equation (4). For simplicity, the values of $n$ and $m$ were chosen to be 1 and 0, respectively.

$$s_i(t) = a_1 s_i(t-1) + b_0 u_i(t) \tag{6}$$

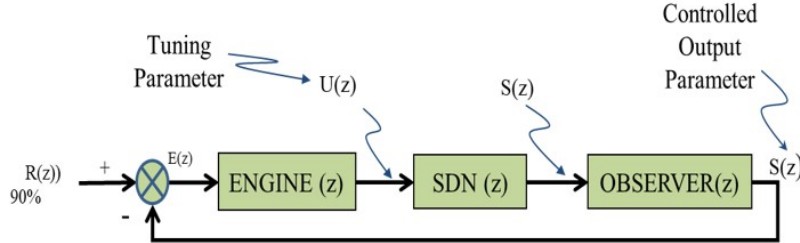

**Figure 3.** Modeling the SFBARMA$_{SDN}$ using feedback system.

### 4.1. SFBARMA$_{SDN}$ Control Law and Gain Design

The SFBARMA$_{SDN}$ engine module used a proportional integral (PI) controller due to its simplicity. SFBARMA$_{SDN}$ used the control law to update the tuning parameter to continuously work on minimizing the current error value in a feedback fashion. The SFBARMA$_{SDN}$ control law is given in Equation (7).

$$u_i(t) = u_i(t-1) + K_i e_i(t-1) \tag{7}$$

The error value, which was fed as an input to the SFBARMA$_{SDN}$ controller, was calculated as a result of the difference between the current reference value and the current level of security level, as shown in Equation (8).

$$e_i(t) = ref_i(t) - s_i(t) \tag{8}$$

In order to be able to compute a stable gain $K$, a root locus was used for the appropriate gain selection. The root locus plot is shown in Figure 4. The gain $K$ was chosen to be 0.5 to obtain the closed loop root of 0.47. Figure 4 shows a plot of the root locus of the system combined with the integral controller. We were seeking a value for the gain K that resulted in poles that were inside the unit circle, causing the closed loop system to be stable. If $K$ resulted in poles that were outside the circle, the system was considered unstable. In fact, when gain $K$ = 0.5 provided stable poles, these were the roots of the root locus.

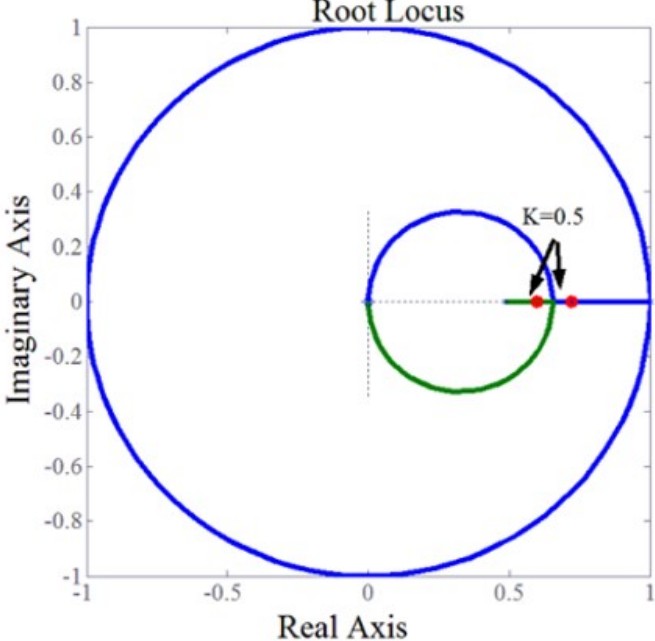

**Figure 4.** Root locus for the SFBARMA$_{SDN}$ model.

### 4.2. SFBARMA$_{SDN}$ Transfer Functions

Transfer functions are mathematical models that express the relationship between the outputs and the inputs. Transfer functions are the frequency domain equivalent for the time domain, as described in Equations (9)–(11). Equation (9) is the Z-transform equivalence for Equation (6).

$$T(z) = \frac{S_i(z)}{U_i(z)} = \frac{b_0 z}{z - a_i} \tag{9}$$

The controller engine module's transfer function is shown in Equation (10). The transfer function carries the control law that relates the tuning parameter's current security level to the error value's tuning parameter.

$$G_i(z) = \frac{U_i(z)}{E_i(z)} = \frac{K_i}{z - 1} \tag{10}$$

To compute the aggregated transfer function of the overall system, the result was obtained by multiplying the transfer functions for each module, resulting in Equation (11).

$$W(z) = \frac{S(z)}{R(z)} = \frac{K \times z \times (z \times b_0)}{(z - 1) \times (z - a_1) + K \times z \times (z - b_0)} \tag{11}$$

SFBARMA$_{SDN}$ used least squares regression to estimate the values of the parameters of the ARMA model $a_1$ and $b_0$. $a_1$ was estimated to be 0.4364, and $b_0$ was estimated to be 0.2897. The goodness of the model was measured using $R^2$. $R^2$ was measured to be 87.5% as an indication of the linearity of the model.

Those transfer functions were used in the Z-transform. The Z- transform was considered as the frequency model of the time domain, which was easier to deal with mathematically. The Z-transform related the output to the input of the system it modeled. The main goal was to design the appropriate gain that resulted in a stable system. The proposed model added feedback that added more control over the desired value of the controlled output parameter using the tuning parameter. The transfer functions shown in Equations (4)–(11) were to design the gain mathematically through the Z-transform mathematical model.

The SFBARMA$_{SDN}$ filtering algorithm started by observing the sequence of the packets to make sure that the data were reliable. If the data were stationary, SFBARMA$_{SDN}$ calculated the *Akaike's information criterion (AIC)*. Packets were filtered; if packets passed the selection criteria, then the packets were considered as *green* packets, otherwise they were considered as *red* packets and discarded. The flowchart for SFBARMA$_{SDN}$ is shown in Figure 5.

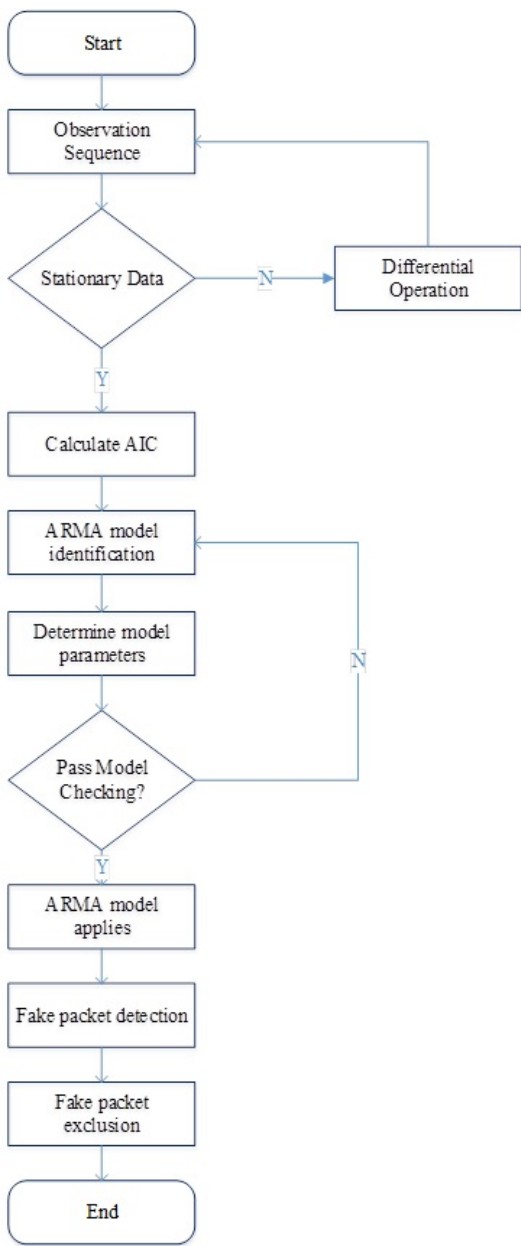

**Figure 5.** SFBARMA$_{\mathrm{SDN}}$ filtering.

## 5. Experimental Testbed and Results

This section provides the experimental testbed followed by the experimental results.

### 5.1. Experimental Testbed

SFBARMA$_{\mathrm{SDN}}$ added a feedback module based on an ARMA regression model that was responsible for adding a security capability to the OpenFlow controller. The added module was responsible for maintaining the required reference value of the security level by computing the overall security in addition to the computation overhead. In order to evaluate the security of the proposed system, a prototype was implemented by adding security capabilities to the existing OpenFlow controller using Python scripts. To simulate attacks, *freedom of information (FOI)* data sets were used. A *Mininet* software emulator tool was installed on a Linux machine. *Wireshark* is a protocol analyzer that was responsible for observing the inbound and outbound packets. The network emulation ran on an HP machine with Intel Core i5-2520M, CPU 2.50 GHz, and RAM 8 GB, running Ubuntu (version 20.04) 64 bits.

In the experimentation setting, attacks varied from 10 to 100 with a step of 10, while the number of packets generated by each attack was set to be 1000. Extensive topologies were used in the experimentation phase. A case study topology that was used for experimentation purposes is shown in Figure 6. The experimental topology was composed of a Ryu SDN controller that was used as an OpenFlow framework. Ryu has high conformance with OpenFlow specifications [15]. The topology consisted of six hosts that were connected to the OpenFlow router. The OpenFlow router was substituted by three types of routers in the experimentation phase. The basic reference OpenFlow, the Bayesian based router SSBN$_{SDN}$, and the secured feedback model based on ARMA. Capabilities were added to secure the OpenFlow controller. Malicious intruders' controllers represented the various malicious attacks.

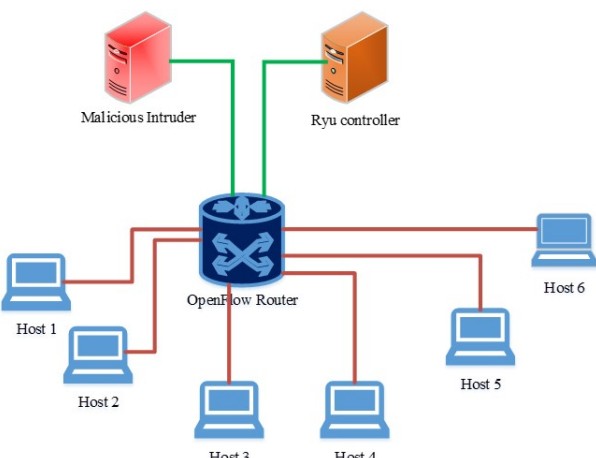

**Figure 6.** Experimental topology.

We tested the new model that was based on feedback and the ARMA model based on the topology in Figure 6. We intend to present more complicated topologies, such as OS3E and EU-GÉANT, in future work.

### 5.2. Experimental Results

In order to evaluate the detection accuracy of the malicious intrusion, SFBARMA$_{SDN}$ added security capabilities to the standard OpenFlow controller. SFBARMA$_{SDN}$ was compared to both the standard OpenFlow controller and also to the SSBN$_{SDN}$. We plotted the tuning parameter versus the controlled output parameter, as shown in Figure 7. We increased the packet load regularly every time slot and measured the effect on the controlled output parameter.

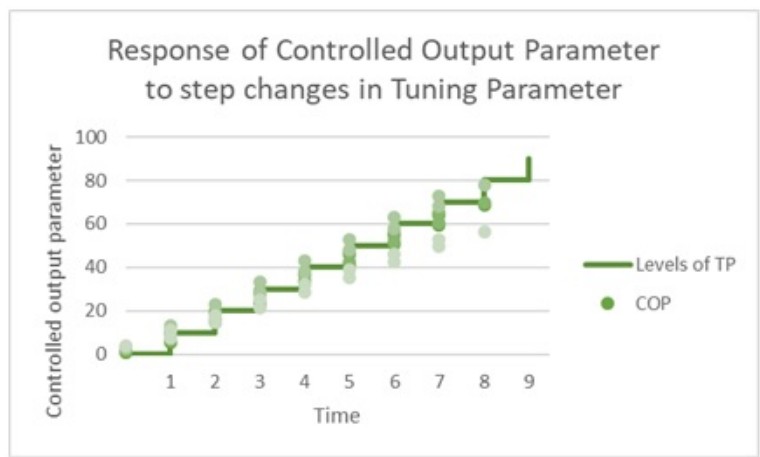

**Figure 7.** Response of the controlled output parameter to step changes in the tuning parameter.

The impact of the tuning parameter on the controlled output parameter clearly suggested that a linear model between the control input and the tuning parameter was observed. We measured the regression factor $R^2$, which represented the goodness of the model. A first order auto regression model was fitted to match data using the linear least squares method where the input to the system was the tuning control parameter and the output was the measurement of the secured system output.

The coefficients of the transfer function for various values of the delay are given in Table 1. The tuning parameter was saturated once the number of incoming packets exceeded the controller capacity. The control saturated at zero because it did not make intuitive sense to have a negative number of fake packets allowed to connect to the system. To help prevent saturation from occurring at the bottom, an integral control scheme was chosen. $\text{SFBARMA}_{\text{SDN}}$ assumed that the success rate to filter fake packets to was set to 90%. Figure 8 compares the number of fake packets for the $\text{SFBARMA}_{\text{SDN}}$ and the two reference models: the standard OpenFlow controller and the $\text{SSBN}_{\text{SDN}}$. The results showed that $\text{SFBARMA}_{\text{SDN}}$ could detect fake packets with a percentage of 91% as opposed to 88% for the $\text{SSBN}_{\text{SDN}}$. Both algorithms outperformed the standard OpenFlow controller as depicted in Figure 8.

**Table 1.** Models coefficient and fits for W(z) transfer function.

| Delay | $R^2$ | $a^1$ | $b^0$ |
| --- | --- | --- | --- |
| 0 | 75.2 | 0.167 | 0.078 |
| 1 | 84.5 | 0.2963 | 0.1193 |
| 2 | 86.2 | 0.3235 | 0.1084 |
| 3 | 87.5 | 0.4364 | 0.2897 |

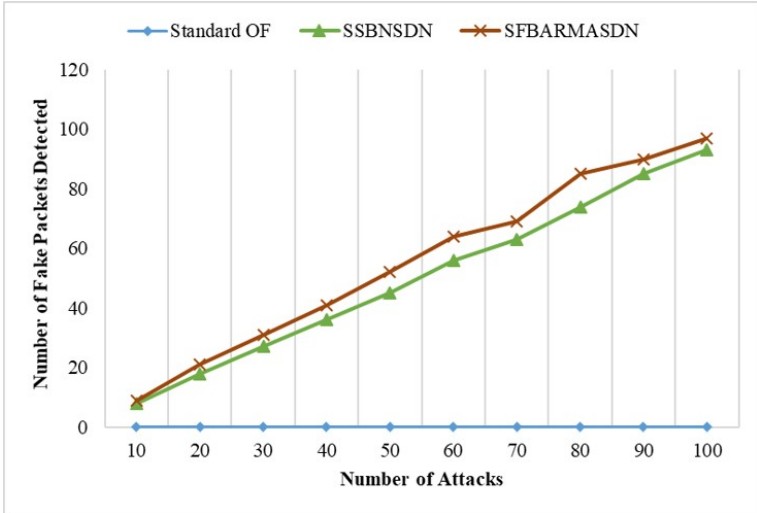

**Figure 8.** Comparison of the fake packets detected for the standard OpenFlow, $\text{SSBN}_{\text{SDN}}$, and $\text{SFBARMA}_{\text{SDN}}$.

Figure 9 shows the comparison of the processing times for the standard OpenFlow, $\text{SSBN}_{\text{SDN}}$ and the proposed $\text{SFBARMA}_{\text{SDN}}$. The experiments showed that the percentage of overhead introduced by the $\text{SFBARMA}_{\text{SDN}}$ when compared to the $\text{SSBN}_{\text{SDN}}$ and the standard OpenFlow was limited to 3% and 5%, respectively. This indicated that the proposed $\text{SFBARMA}_{\text{SDN}}$ did not produce extra overhead when compared to the standard OpenFlow and the $\text{SSBN}_{\text{SDN}}$.

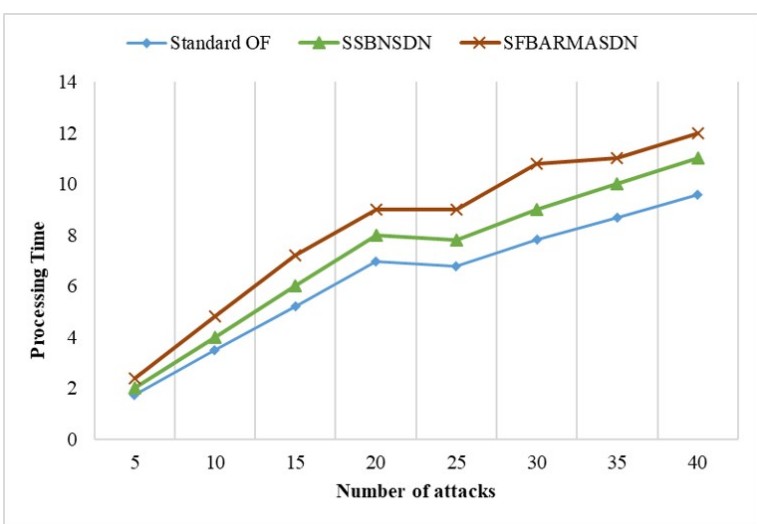

**Figure 9.** Comparison of the processing time among the standard OpenFlow, SSBN$_{SDN}$, and SFBARMA$_{SDN}$.

SFBARMA$_{SDN}$ outperformed both the secured standard using a Bayesian network for SDN (SSBN$_{SDN}$) and the standard OpenFlow in different scenarios by an average improvement of 7% and 80%, respectively. The processing time overhead for the SFBARMA$_{SDN}$ increased by only a percentage of 3% and 5% when compared to the SSBN$_{SDN}$ and the standard OpenFlow, respectively.

## 6. Conclusions and Future Work

This work proposed a novel model to add extra security capability to the standard OpenFlow controllers in SDN networks. The proposed model is referred to as the secured feedback model using autoregressive moving average (ARMA) for SDN networks (SFBARMA$_{SDN}$). SFBARMA$_{SDN}$ added a security capability to the OpenFlow controller through classifying various attacks using packet filters. The filter inspected the properties of the fake packets through feedback control theoretic techniques. SFBARMA$_{SDN}$ used ARMA models to filter fake packets and hence improve the security of the OpenFlow. FBARMA$_{SDN}$ calculated the *Akaike's information criterion (AIC)*. Packets were filtered, if packets passed the selection criteria, then the packets were considered as *green* packets, otherwise they were considered as *red* packets and discarded. In order to measure the added value for the proposed model. The SFBARMA$_{SDN}$ was compared to two reference models. The first reference model was based on Bayesian networking and the second reference model was the standard OpenFlow model. The standard OpenFlow controller optionally implemented transport layer security (TLS).

Extensive experiments were conducted to test the performance of the proposed model. The number of fake packets were measured and compared to two reference models. SFBARMA$_{SDN}$ outperformed both SSBN$_{SDN}$ and the standard OpenFlow in different scenarios by an average value of 7%. The processing time overhead was computed. The processing overhead was around 3% for the SFBARMA$_{SDN}$ compared to the SSBN$_{SDN}$. A virtual network was established using an improved version of the Ryu controller to add the security and filtering capabilities. The results were very promising, SFBARMA$_{SDN}$ outperformed the reference models with minimum overhead.

In future work we intend to use more complicated topologies to test more complex scenarios for the SFBARMA$_{SDN}$. In addition, artificial intelligence-based techniques could also be utilized for filtering. Another research direction for improvement could be to use the ARMA-based models to improve NOX, POX OpenFlow controllers.

**Author Contributions:** Data curation, S.A.; Formal analysis, W.H.F.A.; Methodology, H.K.; Software, N.M. All authors have read and agreed to the published version of the manuscript.

**Funding:** This research received no external funding.

**Conflicts of Interest:** The authors declare no conflict of interest.

## Abbreviations

The following abbreviations are used in this manuscript:

| | |
|---|---|
| ARMA | Auto regressive moving average; |
| BN | Bayesian network; |
| SFBARMA | Secured feedback model using autoregressive moving average; |
| SSBN | Secured standard using a Bayesian network. |

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
