# Peer review of "Feedback ARMA Models versus Bayesian Models towards Securing OpenFlow Controllers for SDNs"

_electronics, doi:10.3390/electronics11091513_

Round 1

Reviewer 1 Report

The overall impression is that the authors fail to emphasize their contributions, and the presented content is a mix of SDN information with some signal processing/filtering theory. Their connection is not clearly explained – significance of signals, etc.

An important part of the theoretical information is common transfer function knowledge [equations (4) – (11)], which should be followed by a more concrete description of the contributions. Moreover, it is not clear what the proposed approach improves in comparison to classical models.

The "Results" section should present more scenarios, topologies, test cases. How is the presented topology/case relevant?

For some parameters, the chosen values do not seem properly explained. For example, what is the relevance of using “the closed loop root of 0.47”? Another example: “a1 is estimated to be 0.4364, and b0 is estimated to be 0.2897.” => not clear how these values are calculated; how are they relevant?

Several issues should be addressed/corrected:

  • Consider removing “Results are very promising.” from the abstract, or integrating it in a more ample/detailed expression. Moreover, exact values for the percentages mentioned in the abstract, such as the performance improvement percentage, should be mentioned and discussed only in the results section, where they can be properly analyzed. Abstract should be only for general description.
  • Missing commas => e.g., in “(…) Internet which (…)”.
  • “The north-bound interfaces is considered as the interface that controls” - use singular form for “interface”; consider removing “is considered as the interface that” and skip directly to “controls”. Apply this kind of corrections in other necessary locations.
  • “the level of vulnerability. and compares the security level” – replace the point with a comma.
  • What is “The goodness of the model” ?

More such examples can be enumerated.

Reviewer 2 Report

In general, I have found this work quite interesting, but there are several aspects that I have not clear.

The Bayesian systems, at least the systems that I have studied, use supervised learning, but in this paper, it is not clear if the authors use a previous phase of supervised learning for the system. If they use this phase, what attack database they have used? They commented that they have used the database freedom of information dataset to simulate the attack, but I have not clear if they have used the same database to training the system. If the authors have used the same database to training and test the system, the result can be overestimated, because the system has been trained specifically for the same attack patter that will suffer.

Other thing that it is not in the paper, it is the parameters used to detect the attack.

Finally, another thing that is a bit lacking in this work, is the data  statistical treatment. A Bayesian system uses probabilities, and when we use probabilities, things like variance, means, confidence interval of the results, number of experiments with different patterns, should be included. My impression is that the authors have only tested with a patter of data.

Round 2

Reviewer 1 Report

Several issues should still be addressed. Examples:

-> Do not use one phrase paragraphs. Merge them into adjacent paragraphs. 

-> Eliminate the line in "-The control saturates at zero because it (...)" => before the capital "T".

-> There is no need to define the Z-transform. The role of the transfer functions in the system it is employed is actually relevant. Also, why are the transfer functions not expressed with negative powers of "z" (causality)?
